# The effectiveness of a Malaysian House Officer (HO) preparatory course for medical graduates on self-perceived confidence and readiness: A quasi-experimental study

**Aneesa Abdul Rashid**[1,2]*, **Sazlina Shariff Ghazali**[1,2], **Iliana Mohamad**[3], **Maliza Mawardi**[1], **Husna Musa**[4], **Dalila Roslan**[5]

**1** Family Medicine Department, Faculty of Medicine and Health Sciences, Universiti Putra Malaysia, Serdang, Malaysia, **2** Malaysian Research Institute on Ageing, Universiti Putra Malaysia, Selangor, Malaysia, **3** Medigrow (Medicorp Resources), Batu Caves, Selangor, Malaysia, **4** Paediatrics Department, Faculty of Medicine and Health Sciences, Universiti Putra Malaysia, Serdang, Malaysia, **5** Kuala Pilah Health District Office, Ministry of Health Malaysia, Negeri Sembilan, Malaysia

* aneesa@upm.edu.my

**Data Availability Statement:** All relevant data are within the manuscript and its Supporting Information files.

## Abstract

### Introduction

House Officer (HO) Preparatory Courses in Malaysia are designed to prepare medical graduates to work as a HO. The courses are designed to address the issues related to lack of confidence and readiness to work, which could lead to stress and HO dropping out of work. The modules focus on how to prepare medical graduates into the real-life working scenario. Hence, we determined the effectiveness of a HO Preparatory Course on the level of confidence and readiness to work among medical graduates.

### Methodology

A quasi-experimental study was conducted at three time-points (pre-intervention, post-intervention and one-month after working as a HO) on the level of confidence and readiness of medical graduates. The intervention was the Medicorp module, which included information and training needed for the HO such as common clinical cases in the wards, case referrals, experience sharing and hands on clinical training. We recruited eligible participants undergoing the course between April–November 2018. The adapted IMU Student Competency Survey was used to measure the confidence and readiness levels, which were scored from a Likert scale of 1–5. The higher score indicated higher levels of confidence or readiness.

### Results

A total of 239 participants were recruited at baseline (90% response rate). They were mostly female (77.8%), Malays (79.1%), single (90.0%), graduated overseas (73.6%), in 2018 (65.3%). The mean (SE) confidence scores significantly increased from 2.18 (1.00) pre-course to 3.50 (0.75) immediately after course and 3.79 (0.92) after one-month of work (p

**Funding:** Universiti Putra Malaysia (UPM) University Community and Transformation Centre (UCTC) (grant no. UPM/UCTC900/3/2/KTGS-05–18). The funders did not play any role in the study design, data collection and analysis, decision to publish, or preparation of the manuscript. URL Link: https://uctc.upm.edu.my/knowledge_transfer_grants/ktgs_jinm_grant-6340.

**Competing interests:** The authors have declared that no competing interests exist.

<0.001, $\eta2 = 0.710$). The mean (SE) readiness scores at pre-course, immediately and one-month post work were 2.36 (1.03), 3.46(0.78) and 3.70(0.90), respectively (p< 0.001, $\eta2 = 0.612$).

## Conclusion

The HO Preparatory Course module was effective in increasing levels of confidence and readiness for medical graduates, most of whom are overseas graduates; namely Egypt, Russia and Indonesia.

## Introduction

University graduates are often said to have lack of confidence and readiness to work [1–3]. A proposed conceptual framework for work readiness skills included people qualities and skills, professional knowledge and skills, and technology knowledge and skills [4]. A study that looked into university graduates in Australia, suggests that there are a lack of synergy between the university curriculum with the development of "work-ready employees"[2]. In the medical line, House Officers (HOs) were reported to feel less confident in working especially in the initial phase of their training [3]. In the UK, around 36%-59% of medical graduates felt that they were prepared to work after one year of graduation [5,6]. Among the factors that make HOs feel unprepared, were the lack of clinical experience they have especially in terms of holding responsibility as a clinician [3]. This is often because of the legal restraint they have during their years as students [3].

There are many factors that affect work readiness among healthcare professionals. These include social intelligence, organisational acumen, work competence and personal characteristics [4]. A qualitative study among nurses and their readiness to work found that education with more "hands-on" experience near the end of the education program was a necessity [7]. Some sectors felt it was essential to create a tool to assess readiness to work in the healthcare industry for example there are validated tools to assess readiness for the physiotherapist [4,8]. Improvement of readiness were successfully seen among surgical interns after implementing modules that used hands on simulation training [9].

The lack of confidence and readiness among HOs may affect their wellbeing. Previous studies reported they were stressed and many were emotionally burnt out [10–12]. This can lead to extension of their posting, and sometimes even dropping out of the HO training all together. This gives a negative impact on not just the healthcare system, but also the economy [13]. With this in mind, a few steps have been taken by some organisations in Malaysia to organise HO Preparatory courses. These courses are aimed at medical graduates to prepare themselves for the HO postings. These courses have become increasingly popular and are opened to all medical graduates who plan to work in Malaysia. It focuses on preparing the participants to cope with the needs of the working environment. This includes information on the technical details of the working shift system and on call system, and knowledge on the work scope such as clerking, referrals and ward management. In addition, soft skills such as time management is also included. This course is carried out during the waiting period, between the time they graduate until the time they officially work as a HO [13]. This period would range between 6–15 months. However, there are limited published studies that evaluates the effectiveness of these courses. Therefore, we assessed the effectiveness of a HO Preparatory course in addressing medical graduates' confidence and readiness to work.

## Materials and method

This was a pre-post, quasi-experimental study evaluating the effectiveness of a HO Preparatory module intervention on medical graduates. The participants received an intervention which was the HO Preparatory course. The participants were required to answer a self-administered pre-tested structured questionnaire prior to intervention and immediately post intervention. At the one-month after working as a HO follow-up, the participants were interviewed via a telephone call guided by the pre-tested structured questionnaire. The participants were initially reminded of their participation on this study via WhatsApp messages, which was part of the effort to facilitate the module a few days before making the call. The participants were then followed up via calls after office hours. If there was no response, another two reminders via personal messages was sent approximately one week apart. Participants would reply the messages on which the appropriate dates were available to be interviewed. There was no control group due to limitation of time and resources.

### Setting and sample

Participants involved in the study were medical graduates who registered to attend the HO Preparatory course from April until November 2018. Those who had not graduated, already working as a HO and with known diagnosis of psychiatric illness were excluded from this study. The site of this study was at an International Youth Centre (IYC) near the capital city of Kuala Lumpur. The location was central to allow ease for participants from other states to attend the program. The IYC was equipped with training and boarding facilities.

The sample size was calculated using the G*Power 3.1 sample size calculator software using mean confidence levels of HOs in a study by Williams et al [14]. After accounting for a 30% attrition rate, 80% power and 0.05% significance level, the estimated sample size was 208.

The participants were informed regarding the study by the researchers prior to starting the HO Preparatory course. Verbal and written informed consents were taken from the eligible participants who agreed to participate in the study prior to enrolling in the course. Privacy and confidentiality were ensured and maintained. Ethical approval was obtained from the Ethics Committee Involving Human Subjects Universiti Putra Malaysia (JKEUPM-2018-054) and the Medical Research Ethics Committee, Ministry of Health Malaysia (NMRR-18-978-41224). The study protocol was registered with the National Institutes of Health as trial registration (NCT03510195).

### The HO preparatory course module intervention

As mentioned earlier there are many HO preparatory courses available, but not many of them are held on a consistent and regular basis. Medicorp has been conducting HO Preparatory courses consistently for the past two years, on a monthly basis. Prior to this, the HO Preparatory course was conducted by a medical non-governmental organization since 2012. It was later on privatised. It is a three-day program that relies heavily on feedback of its alumni, many of whom are now medical officers and young specialists. The course charges RM 450 (USD 100) for a three-day course this includes cost of venue, refreshments and food, training facilities such as medical equipment and mannequins, and trainer fees. The cost of running this HO preparatory course is around RM 350–400 per person. Members of the alumni comes back as facilitators and trainers to help with the training. Also, at the same time they would give feedback on how to improve the module. Involvement of the alumni as peer trainers remain one of the unique qualities of Medicorp. Often times the senior HOs would become the trainers and give tips and tricks to help out the juniors. Apart from that, Medicorp has established an exclusive follow-up system starting from before the course, where participants

are enlisted in a WhatsApp group. The participants are later followed up after completion of the course to assist them with the HO application process which are guided by Medicorp's staff. Malaysia has an online system for HO placements, which is called the e-Houseman. This online portal is opened for registration at specific times in the year. This process is not found in other HO preparatory courses. Details of the course has been described elsewhere [13].

### Outcome measures

The primary outcome measured in this study was confidence to work as a HO, while the secondary measure was readiness to work as a HO and their psychological well-being.

Confidence and readiness were measured using the adapted International Medical University (IMU) Student Competency Survey [15–17]. The adaptation of this survey has been discussed elsewhere [13]. The participants were asked on how confident they are in practical skills, generic skills, and personal skills. They rated their confidence level on a Likert scale of 1 to 5 with the higher scores representing higher levels of confidence. The level of readiness had only one question, asking the participants how ready they are should they have to begin work tomorrow. The same Likert scale was used, asking the participants to rate their readiness levels on a scale of 1 to 5. The adapted questionnaire was pretested, and the subscales had Cronbach alpha ranging between 0.92 and 0.96. The original authors of the tool used to assess confidence and readiness among senior students, had decided the level of confidence to be a minimum level of '3' for their final year students. This was the standard set in a local Malaysian University [15]. We did not disclose this to the participants as we did not want this information to effect their response to the questionnaire.

In addition, the psychosocial wellbeing of the participants were measured using the Depression, Anxiety and Stress Scale with 21 items (DASS-21). It is a valid and reliable tool with a Cronbach's alpha of 0.96 to 0.97 for DASS-Depression, 0.84 to 0.92 for DASS-Anxiety, and 0.90 to 0.95 for DASS-Stress [18]. In this scale, the higher the score, the higher the levels of depression, anxiety and stress. The normal scores are 0–9, 0–7, 0–14 respectively. Participants with abnormal scores would be contacted via the follow up system and were advised who and where to seek help.

### Statistical analysis

Data was analysed using the IBM Social Package for Social Science (SPSS) V.24. A descriptive analysis of participant demographic characteristics, clinical experience and baseline level of confidence, readiness and psychological well-being were presented as mean and standard deviation (SD) for continuous variables and as frequency and percentage for categorical variables. Chi-square or Fisher's exact tests were conducted to compare between participants who completed and withdrew from the study. A repeated measures analysis of variance was conducted to determine intervention effectiveness within the groups from baseline, immediately after the intervention to one-month after starting work. The results were presented as mean and standard error (SE), partial eta squared, and significant value was set at $p < 0.05$.

### Results

A total of 267 participants registered for the course and were invited to participate within the timeframe of this study period of recruitment. Only 239 returned their baseline questionnaires. At the time-point immediately after the intervention, 224 participants returned their completed questionnaires. At the subsequent follow up of the study, which was one-month post working as a HO, 101 answered the questionnaire. Fig 1 illustrates the flow of study participants in this study. Comparison of baseline characteristics of the participants who were lost to

**Fig 1. Flow of study participants.**

follow-up and those who completed the study showed no difference in their age (p = 0.322), gender (p = 0.739), ethnicity (p = 0.931), marital status (p = 0.499), levels of overall confidence (p = 0.800) and readiness to work (p = 0.141), DASS-Depression level(p = 0.340), DASS-Anxiety level (p = 0.513) and DASS-Stress level (p = 0.076) (results not shown).

Table 1 shows the sociodemographic details of the participants. Most of the participants were aged 25.66 (1.54) years old. They were mainly females (186, 77.8%), not married (215, 90%), Malay (189, 79.1%), Muslims (194, 81.2%), graduated from outside of a Malaysian institution (176, 73.6%) in 2018 (156, 65.3%).

The mean confidence level for practical tasks, generic and personal skills over time (at baseline, immediately after (post) intervention and one-month after working as a HO) are shown in Table 2. The levels of confidence were significantly increased across all the three domains over time except for "taking a history and performing relevant examination at first assessment

**Table 1. Sociodemographic factors of participants (N = 239).**

| Factors | n (%) |
|---|---|
| **Age,** mean (SD) (years) | 25.66 (1.54) |
| **Gender** | |
| Male | 53 (22.2) |
| Female | 186 (77.8) |
| **Marital Status** | |
| Single/ never married | 215 (90.0) |
| Married | 24 (10.0) |
| **Ethnicity** | |
| Malay | 189 (79.2) |
| Chinese | 19 (7.9) |
| Indian | 19 (7.9) |
| Others | 12 (5.0) |
| **Religion** | |
| Islam | 194 (81.2) |
| Christian | 18 (7.5) |
| Hindu | 16(6.7) |
| Buddha | 7 (2.9) |
| Others | 4 (1.7) |
| **Year of Graduation** | |
| 2016 | 2 (0.8) |
| 2017 | 81 (33.9) |
| 2018 | 156 (65.3) |
| **Place of Graduation** | |
| Local | 63 (26.4) |
| Overseas | 176 (73.6) |

SD = standard deviation.

of new admissions" in the generic skills sub-domain, and "lumbar puncture" in the practical tasks sub-domain when comparing the baseline scores to one-month post work scores.

There was a significant increase for overall confidence and readiness to start working as a HO when comparing baseline mean scores with immediately after intervention and one-month post work. Mean scores significantly increased at immediately post intervention 3.50 (0.75) and 3.46 (0.78) for both confidence and readiness scores, respectively. Table 3 summarises the mean score for the overall confidence and readiness.

Table 4 shows the mean scores for depression, anxiety and stress among the participants. All the mean scores are within the normal range. There was a decrease in the levels of depression, anxiety and stress after one-month of work. There was a significant reduction of levels of stress (p = 0.01) from 11.78 (8.05) to 8.58 (10.33) one-month after working as a HO. The normal stress scores are 0–14.

## Discussion

This study found the HO Preparatory course to be effective in increasing the level of confidence and readiness among medical graduates immediately after and one-month after working, compared to baseline. We also report the levels of depression, anxiety and stress reduced

**Table 2. Mean confidence level for practical tasks, generic and personal skills over time.**

| | Mean confidence level (SE) at baseline | Mean confidence level (SE) at post intervention | P-value | Mean confidence level (SE) at one-month after working | P-value | Partial η2 |
|---|---|---|---|---|---|---|
| **Generic Skills** | | | | | | |
| Taking a history and performing relevant examination at first assessment of new admissions | 3.29(3.19) | 3.55(0.88) | 1.00 | 3.99(0.72) | 0.097 | 0.187 |
| Make plan of management for new admissions | 2.23(0.97) | 3.32(0.84) | <**0.001**** | 3.52(0.80) | <**0.001**** | 0.614 |
| Recognizing sick patients | 2.96(0.92) | 3.77(0.67) | | 3.79(0.75) | | 0.506 |
| Functioning as a team member in assessing and managing sick patients | 2.87(0.91) | 3.85(0.68) | | 3.83(0.80) | | 0.564 |
| Prioritizing and managing ward work | 2.44(1.04) | 3.83(0.80) | | 3.88(0.78) | | 0.686 |
| **Practical tasks** | | | | | | |
| Starting resuscitation in hospital | 2.22(0.78) | 3.69(0.77) | <**0.001**** | 2.90(1.01) | <**0.001**** | 0.738 |
| IV-line insertion (adult) | 2.82(1.02) | 4.06(0.80) | | 4.19(0.79) | | 0.656 |
| Blood taking (adult) | 3.18(1.11) | 4.25(0.72) | | 4.34(0.73) | | 0.562 |
| Inserting urinary catheter (male) | 2.79(1.14) | 4.33(0.67) | | 4.07(0.96) | | 0.637 |
| Inserting urinary catheter (female) | 2.76(1.16) | 4.31(0.66) | | 4.29(0.81) | | 0.650 |
| Do basic suturing and tie | 2.49(0.95) | 3.93(0.81) | | 3.40(1.00) | | 0.720 |
| Prescribing common medications (format, not dosage) | 2.31 (0.93) | 3.33(0.91) | | 4.11(0.82) | | 0.703 |
| | **Mean confidence level (SE) at baseline** | **Mean confidence level (SE) at post intervention** | **P-value** | **Mean confidence level (SE) at one-month after working** | **P-value** | **Partial η2** |
| Requesting radiological investigations like CXR, CT | 2.30(0.95) | 3.64(0.80) | | 3.95(0.97) | | 0.717 |
| Do a comprehensive review on patients during rounds | 2.14(0.84) | 3.82(3.04) | | 3.93(0.83) | | 0.748 |
| Referring cases to another department | 1.99(0.83) | 3.60(0.74) | | 3.73(0.90) | | 0.785 |
| Assisting operations | 2.09(0.90) | 3.35(0.84) | | 3.66(0.98) | | 0.724 |
| Prescribing IV fluid (format of writing) | 1.90(0.80) | 3.15(0.91) | | 3.64(0.85) | | 0.742 |
| Lumbar Puncture | 1.67(0.74) | 2.29(1.13) | | 1.89(0.88) | 0.577 | 0.264 |
| **Personal skills** | | | | | | |
| Team-working: e.g. sharing ward work, arranging rosters | 3.32(1.09) | 3.96(0.71) | <**0.001**** | 4.04(0.85) | <**0.001**** | 0.269 |
| Handling criticisms from your senior colleagues | 3.04(1.10) | 3.78(0.82) | | 4.02(0.93) | | 0.413 |
| Coping with additional, unexpected tasks | 3.03(0.10) | 3.72(0.81) | | 3.88(0.98) | | 0.338 |
| Working independently away from home | 3.39(1.06) | 3.68(0.97) | **0.042*** | 3.98(0.99) | | 0.159 |
| Referring cases to seniors | 2.74(1.00) | 3.70(0.76) | <**0.001**** | 4.06(0.81) | | 0.565 |

SE = Standard error

*p < 0.05 &

**p<0.001 = statistical significance.

one-month after work compared to baseline. However, only reduction of the stress levels was significant.

We found all confidence levels in practical task and generic skills pre-intervention lower compared to a local Malaysian university final year medical students in a previous study [15]. This could be contributed by the large number of participants from overseas whom are not exposed to clinical procedures due the strict policies in the countries they studied [19]. Another reason for this may be because local graduates are more familiar with the Malaysian

**Table 3. Overall mean confidence and readiness scores over time.**

|  | At baseline | At post intervention | P-value | At one-month after working | P-value | Partial η2 |
|---|---|---|---|---|---|---|
| Mean (SE) Overall confidence score | 2.18(1.00) | 3.50(0.75) | <**0.001**** | 3.79(0.92) | <**0.001**** | 0.710 |
| Mean (SE) Readiness score | 2.36(1.03) | 3.46(0.78) |  | - | - | - |

SE = standard error

**p<0.01 = statistical significance.

hospital setting and the common illnesses encountered in the country. In addition to that, the syllabus in overseas institutions may not be consistent with the standard Malaysian syllabus [19]. We also note that the majority of participants were female. This suggests more females feel less confident and ready to start work. Male doctors have been reported to being more confident when it comes to working. Females, despite displaying the same level of performance have been reported to have lesser confidence and are more anxious [20,21].

Post intervention, confidence levels reached the minimum amount required for final year medical students which was a score of '3' (except for 'performing a lumbar puncture') [15,16]. Both results for immediately post intervention and one-month after work showed a significant increase in confidence levels. For work readiness, our scores were comparable to a study done on final year medical students who were towards the end of training at a local university [17]. Another study done in Croatia reports that their participants scored 5 out of a possible 10 score for their readiness to work with patients. This is assumed to be lower as the participants in this post intervention group scored slightly higher than half the total score of 5 [22]. However, this study did not have any interventions done on participants.

There has been several studies looking into the preparedness of junior doctors to commence work. In a previous qualitative study in the United Kingdom (UK) researchers found that lack of preparedness among medical graduates were because many of them were merely observers during their student postings and did not play an active role as a team [23]. While another study in Australia suggests that certain types of training may be carried out jointly with the universities and hospitals [24]. Though these may include policies on a bigger platform level involving the stakeholders, the HO Preparatory courses may be a small step towards addressing the issues of HO confidence and readiness to work. This is because it addresses the medical graduates' concerns in a different perspective. The course is a compilation of didactic learning and simulation, helping participants understand the scope of work and what is expected of them in the local setting. This is similar to a surgical intern course carried out in the United States (US) that was found to be an effective course [9]. Apart from that, the method of learning from peers and seniors are a favourable method among doctors [25]. This is incorporated in the HO Preparatory courses making it an attractive method of learning.

Focusing on psychosocial wellbeing, this study found the mean level of depression, stress and anxiety were within normal range. Although we noted the anxiety level to be slightly

**Table 4. Overall mean scores for depression, anxiety and stress.**

|  | Mean (SD) | Mean post one-month working (SD) | p | Partial η2 |
|---|---|---|---|---|
| Depression | 8.67 (7.93) | 7.80(8.95) | 0.479 | 0.005 |
| Anxiety | 9.75(7.24) | 7.43(9.54) | 0.063 | 0.035 |
| Stress | 11.78 (8.05) | 8.58(10.33) | **0.019*** | 0.055 |

*p<0.05 = statistical significance.

higher prior to intervention with a score 9.75 (normal range: 0–9). It has been reported that the prevalence of anxiety among HOs are similarly high in other local centres which are around 60% - 64% [11,12]. A previous study reported that the high level of anxiety among HOs were associated with them perceiving as being bullied [11]. On the other hand, those that are stressed and depressed are more likely to think of quitting [11]. The levels of stress had significantly lowered in this study. Hence the intervention has proven effective in addressing the needs of the participants when dealing with stressors in their working environment.

## Strength and limitation

This is the first study of its kind evaluating the effectiveness of a HO Preparatory course that specifically looks into medical graduates as a participant, and following them up after they start work as a HO. However, there are some limitations of this present study. Firstly, it did not have a control, therefore we were unable to compare with those that did not receive intervention. This study is not representative of medical graduates in Malaysia as a whole, as this course is not compulsory and is opened to those who wish to join. Hence, attracting only those who feel they needed help in preparing for their HO-ship. It is also important to mention that this is a paid course, thus limiting the participants to those that have the financial means. Lastly, there are many factors that may affect the confidence, readiness and psychological well-being post working one-month including hospital HO inductions and the social support of the participants. The most challenging part of the study was following up the participants when they have started working due to their busy schedules. The other difficulty encountered during this study that contributed to the low rate of retention of participation towards the end of the study, was obtaining consent from the individual hospitals the participating HOs worked. This was done to fulfil the criteria of the ethics board.

## Conclusion and recommendations

This HO Preparatory Course module, involving mostly overseas medical graduates was effective in increasing levels of confidence and readiness to work as a HO. In addition, there was a significant reduction of stress levels among the participants compared to before the intervention. There is a need for more robust future studies that compares a HO Preparatory course with controls. Furthermore, the needs and concerns of medical graduates into preparation for working life should be explored in future studies through qualitative methods.

## Supporting information

**S1 Table. Mean confidence levels for generic, practical task and personal skills at different time points.**
(DOCX)

**S2 Table. Overall mean confidence and readiness scores at different time points.**
(DOCX)

**S1 File. Questionnaire.**
(PDF)

**S2 File. Quasi-experimental study on the effectiveness of a house officer preparatory course for medical graduates on self-perceived confidence and readiness: A study protocol.**
(PDF)

## Acknowledgments

The authors would like to thank the funders of this study, Universiti Putra Malaysia (UPM) University Community and Transformation Centre (UCTC) (grant no. UPM/UCTC900/3/2/ KTGS-05–18) and Medigrow (Medicorp Resources) for their collaboration in research planning and training. We thank Dr Halidah Mohd Yusuf, our research assistant in assisting this project. We would also like to thank the Director General of Health Malaysia for his permission to publish this article.

## Author Contributions

**Conceptualization:** Aneesa Abdul Rashid, Sazlina Shariff Ghazali, Iliana Mohamad, Maliza Mawardi, Husna Musa, Dalila Roslan.

**Data curation:** Aneesa Abdul Rashid, Sazlina Shariff Ghazali, Maliza Mawardi.

**Formal analysis:** Aneesa Abdul Rashid, Sazlina Shariff Ghazali, Maliza Mawardi.

**Funding acquisition:** Aneesa Abdul Rashid, Sazlina Shariff Ghazali.

**Investigation:** Aneesa Abdul Rashid, Sazlina Shariff Ghazali.

**Methodology:** Aneesa Abdul Rashid, Sazlina Shariff Ghazali, Iliana Mohamad, Husna Musa, Dalila Roslan.

**Project administration:** Aneesa Abdul Rashid, Iliana Mohamad, Husna Musa, Dalila Roslan.

**Resources:** Aneesa Abdul Rashid, Iliana Mohamad.

**Supervision:** Aneesa Abdul Rashid, Sazlina Shariff Ghazali.

**Validation:** Maliza Mawardi.

**Writing – original draft:** Aneesa Abdul Rashid, Sazlina Shariff Ghazali.

**Writing – review & editing:** Aneesa Abdul Rashid, Sazlina Shariff Ghazali, Iliana Mohamad, Maliza Mawardi, Husna Musa, Dalila Roslan.

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
