## [Decision Letter · Decision Letter 0]

13 May 2020

PONE-D-19-35709

The Effectiveness of a Malaysian House Officer (HO) Preparatory Course for Medical Graduates on Self-Perceived Confidence and Readiness: A Quasi-Experimental Study The Effectiveness of a HO Preparatory Course

PLOS ONE

Dear Dr Rashid,

Thank you for submitting your manuscript to PLOS ONE. After careful consideration, we feel that it has merit but does not fully meet PLOS ONE’s publication criteria as it currently stands. Therefore, we invite you to submit a revised version of the manuscript that addresses the points raised during the review process.

I am recommending minor revision, please submit point by point response to address the reviewers comments. 

We would appreciate receiving your revised manuscript by Jun 27 2020 11:59PM. To enhance the reproducibility of your results, we recommend that if applicable you deposit your laboratory protocols in protocols.io, where a protocol can be assigned its own identifier (DOI) such that it can be cited independently in the future. For instructions see: http://journals.plos.org/plosone/s/submission-guidelines#loc-laboratory-protocols

We look forward to receiving your revised manuscript.

Kind regards,

Charles A. Ameh, PhD, MPH, FWACS (OBGYN), FRCOG

Academic Editor

PLOS ONE

Journal Requirements:

2. PLOS ONE will consider submissions that present new methods, software, or databases as the primary focus of the manuscript if they meet the criteria of utility, validation, and availability described here: http://journals.plos.org/plosone/s/submission-guidelines#loc-methods-software-databases-and-tools. To meet these criteria, please provide supporting materials enabling other teachers and researchers to replicate your teaching intervention such as sample worksheets, a detailed lesson plan or curriculum or other educational materials. If you include supporting materials, they should not be under a copyright more restrictive than CC-BY.

4. We note you have included a table to which you do not refer in the text of your manuscript. Please ensure that you refer to Table 3 in your text; if accepted, production will need this reference to link the reader to the Table.

5. Please include your tables as part of your main manuscript and remove the individual files. Please note that supplementary tables (should remain/ be uploaded) as separate "supporting information" files

Additional Editor Comments (if provided):

Thanks for your submission, I am recommending minor revision, please submit point by point response to address the reviewers comments.

Reviewers' comments:

Reviewer's Responses to Questions

**Comments to the Author**

1. Is the manuscript technically sound, and do the data support the conclusions?

Reviewer #1: Yes

Reviewer #2: Yes

2. Has the statistical analysis been performed appropriately and rigorously? 

Reviewer #1: I Don't Know

Reviewer #2: Yes

3. Have the authors made all data underlying the findings in their manuscript fully available?

Reviewer #1: No

Reviewer #2: Yes

4. Is the manuscript presented in an intelligible fashion and written in standard English?

Reviewer #1: No

Reviewer #2: Yes

5. Review Comments to the Author

Reviewer #1: This study explores the effectiveness of a HO preparation course conducted for newly qualified doctors in Malaysia, comparing confidence, readiness for practice and psychological well-being.

On the whole the findings seem to support the authors conclusions although they rightly include in the limitations of the study, the lack of a control group and the large loss to follow-up at 1 month post employment. The study includes a high proportion of the participants who qualified overseas (although it is not stated where) which it may be worth reflecting in their conclusions.

- The authors state they can not publish the data due to personal information it contains. Is it not possible to publish an anonymised version of the data set, as happens with many other studies?

- The use of English could be improved generally but particularly in lines 103-104 and 280-281.

- The table attribution in line 241 is incorrect.

- In line 272, the authors describe a minimum confidence level require for final year students. It would be beneficial to clarify this point including who sets the minimum, are the students/newly qualified doctors aware of this?

- There is not mention of the cost of the course, just that it is now a private enterprise. Can they comment on the cost of doing the course and any impact this might have on the recruitment or findings of the current study.

Reviewer #2: This is an interesting and well written paper that makes a valuable contribution. I would be interested to know if House Officers were provided with a formal induction by the hospitals they worked in and if so what the nature of the induction was as this may have had a bearing on the scores one month after commencing work.

The paper would also be improved by providing a more detailed description as to how the skills were taught, for example, were models used to teach practical skills. The topics covered seem a lot for a three day course and it would be interesting to understand a bit more as to the timetable and structure of the course.

I am concerned about making direct comparisons with the 1 month after starting score as the numbers returning the survey were less than half. But nevertheless this paper makes a meaningful contribution to an important subject

6. PLOS authors have the option to publish the peer review history of their article (what does this mean?). If published, this will include your full peer review and any attached files.

Reviewer #1: No

Reviewer #2: No

---

## [Author Response · Author response to Decision Letter 0]

29 May 2020

Dear Reviewers and Editors, 

Thank you for the constructive comments in making this manuscript better. We have answered the comments as best we could below. We have also attached a table for easier viewing to address each comment in the "comments for reviewer" file. 

Editor & Journal Requirements

To enhance the reproducibility of your results, we recommend that if applicable you deposit your laboratory protocols in protocols.io, where a protocol can be assigned its own identifier (DOI) such that it can be cited independently in the future. For instructions see: http://journals.plos.org/plosone/s/submission-guidelines#loc-laboratory-protocols

>> Thank you for your suggestion, however our protocol is already published at: https://bmjopen.bmj.com/content/9/8/e024488

It is registered in https://bmjopen.bmj.com/content/9/8/e024488

It is also attached as S2 File

>> We have done our best to ensure that the amended manuscript meet the PLOS ONE’s style

2. PLOS ONE will consider submissions that present new methods, software, or databases as the primary focus of the manuscript if they meet the criteria of utility, validation, and availability described here: http://journals.plos.org/plosone/s/submission-guidelines#loc-methods-software-databases-and-tools. To meet these criteria, please provide supporting materials enabling other teachers and researchers to replicate your teaching intervention such as sample worksheets, a detailed lesson plan or curriculum or other educational materials. If you include supporting materials, they should not be under a copyright more restrictive than CC-BY.

>>Thank you for your suggestions. We attach our questionnaire (S1 file), and also the details of the module and timeline of the course is published in the protocol paper as mentioned above (S2 file)

>> Thank you for highlighting about the data availability as required by PLOS One. We have included 2 supporting tables to describe the data we used in this study (S1 & S2 Tables). However, we did not put these supporting tables in our manuscript. At present, we are not able to share the raw data as we are doing more ongoing data analysis for upcoming publications related to this research project. 

>> Answered as above

4. We note you have included a table to which you do not refer in the text of your manuscript. Please ensure that you refer to Table 3 in your text; if accepted, production will need this reference to link the reader to the Table.

>> We have made the relevant corrections, thank you for pointing out the error in type.

5. Please include your tables as part of your main manuscript and remove the individual files. Please note that supplementary tables (should remain/ be uploaded) as separate "supporting information" files

>> We have done as per requested

Reviewer 1

This study explores the effectiveness of a HO preparation course conducted for newly qualified doctors in Malaysia, comparing confidence, readiness for practice and psychological well-being.

>>Thank you for your kind comments, we have tried to address all your comments as best we could

On the whole the findings seem to support the authors conclusions although they rightly include in the limitations of the study, the lack of a control group and the large loss to follow-up at 1 month post-employment. 

>>It was most challenging to follow up the young doctors as mentioned in our limitations, this included the busy schedule and also the large amount of administrative work of following up each hospital to request permission on follow up of the participants. 

The study includes a high proportion of the participants who qualified overseas (although it is not stated where) which it may be worth reflecting in their conclusions.

>> Thank you for this suggestion, we have added on this to conclusion and in the abstract

- The authors state they can not publish the data due to personal information it contains. Is it not possible to publish an anonymised version of the data set, as happens with many other studies?

>> Thank you for highlighting about the data availability as required by PLOS One. We have included 2 supporting tables to describe the data we used in this study (S1 & S2 Tables). However, we did not put these supporting tables in our manuscript. At present, we are not able to share the raw data as we are doing more ongoing data analysis for upcoming publications related to this research project.

- The use of English could be improved generally but particularly in lines 103-104 and 280-281.

>> We have corrected the grammar on the mentioned lines and also the article as a whole

- The table attribution in line 241 is incorrect.

>> We have corrected it and attributed table 3 and 4 in their rightful position

- In line 272, the authors describe a minimum confidence level require for final year students. It would be beneficial to clarify this point including who sets the minimum, are the students/newly qualified doctors aware of this?

>> The original authors of the tool used to assess confidence and readiness among senior students, had decided the level of confidence to be a minimum level of ‘3’ for their final year students. This was the standard set in a local Malaysian University [15]. We did not disclose this to the participants as we did not want this information to effect their response to the questionnaire.

- There is not mention of the cost of the course, just that it is now a private enterprise. 

- Can they comment on the cost of doing the course and any impact this might have on the recruitment or findings of the current study.

>>Thank you for this comment, we agree on this statement and have included explanations as below:

Materials & Methods/ outcome measures:

The course charges RM 450 (USD 100) for a three-day course this includes cost of venue, refreshments and food, training facilities such as medical equipment and mannequins, and trainer fees. The cost of running this HO preparatory course is around RM 350-400 per person

Strengths and limitations:

It is also important to mention that this is a paid course, thus limiting the participants to those that have the financial means

Reviewer 2

This is an interesting and well written paper that makes a valuable contribution. 

>> Thank you for the encouraging comment, we feel that interventions such as this which is highly popular here, needs an evidence-based assessment

I would be interested to know if House Officers were provided with a formal induction by the hospitals they worked in and if so what the nature of the induction was as this may have had a bearing on the scores one month after commencing work.

>> From the feedback of the alumni, all hospitals would have their formal induction for house officers but is mainly on administrative issues and not on the hands-on experience itself. 

We do acknowledge this as one of the potential bias of the follow up cohort and can be considered in future experiments which we recommend looking into control groups (mentioned in conclusion and recommendation)

We have also added this point in ‘strength and limitations’: 

Lastly, there are many factors that may affect the confidence, readiness and psychological well-being post working one-month including hospital HO inductions and the social support of the participants

The paper would also be improved by providing a more detailed description as to how the skills were taught, for example, were models used to teach practical skills. The topics covered seem a lot for a three day course and it would be interesting to understand a bit more as to the timetable and structure of the course.

>> Thank you for the comment, the details of this course is published in a paper entitled Quasi-experimental study on the effectiveness of a house officer preparatory course for medical graduates on self-perceived confidence and readiness: a study protocol 

We have attached this as file as additional file S1 for your reference

I am concerned about making direct comparisons with the 1 month after starting score as the numbers returning the survey were less than half. 

>> We encountered many challenges in the follow up of this experiment. We did our best and invested tremendous efforts to follow up the participants in which we mentioned in ‘material and methods’

The participants were initially reminded of their participation on this study via WhatsApp messages, which was part of the effort to facilitate the module a few days before making the call. The participants were then followed up via calls after office hours. If there was no response, another two reminders via personal messages was sent approximately one week apart. Participants would reply the messages on which the appropriate dates were available to be interviewed.

But nevertheless this paper makes a meaningful contribution to an important subject

>> Thank you for your encouraging remark, we hope that this will help facilitate future efforts to improve medical training in the near future.

Thank you

Yours sincerely

Dr Aneesa Abdul Rashid

---

## [Decision Letter · Decision Letter 1]

22 Jun 2020

The Effectiveness of a Malaysian House Officer (HO) Preparatory Course for Medical Graduates on Self-Perceived Confidence and Readiness: A Quasi-Experimental Study The Effectiveness of a HO Preparatory Course

PONE-D-19-35709R1

Dear Dr. Rashid,

We’re pleased to inform you that your manuscript has been judged scientifically suitable for publication and will be formally accepted for publication once it meets all outstanding technical requirements.

Kind regards,

Charles A. Ameh, PhD, MPH, FWACS (OBGYN), FRCOG

Academic Editor

PLOS ONE

Additional Editor Comments (optional):

Congratulations for addressing all the comments. I am pleased to recommend your manuscript for publication.

Reviewers' comments:

Reviewer's Responses to Questions

**Comments to the Author**

1. If the authors have adequately addressed your comments raised in a previous round of review and you feel that this manuscript is now acceptable for publication, you may indicate that here to bypass the “Comments to the Author” section, enter your conflict of interest statement in the “Confidential to Editor” section, and submit your "Accept" recommendation.

Reviewer #1: All comments have been addressed

Reviewer #2: All comments have been addressed

2. Is the manuscript technically sound, and do the data support the conclusions?

Reviewer #1: Yes

Reviewer #2: Yes

3. Has the statistical analysis been performed appropriately and rigorously? 

Reviewer #1: I Don't Know

Reviewer #2: Yes

4. Have the authors made all data underlying the findings in their manuscript fully available?

Reviewer #1: Yes

Reviewer #2: Yes

5. Is the manuscript presented in an intelligible fashion and written in standard English?

Reviewer #1: Yes

Reviewer #2: Yes

6. Review Comments to the Author

Reviewer #1: (No Response)

Reviewer #2: The manuscript is now ready for publication in my view. Issues raided by reviewers have been addressed and revisions made.

7. PLOS authors have the option to publish the peer review history of their article (what does this mean?). If published, this will include your full peer review and any attached files.

Reviewer #1: No

Reviewer #2: No

---

## [Editor Report · Acceptance letter]

26 Jun 2020

PONE-D-19-35709R1 

The Effectiveness of a Malaysian House Officer (HO) Preparatory Course for Medical Graduates on Self-Perceived Confidence and Readiness: A Quasi-Experimental Study The Effectiveness of a HO Preparatory Course 

Dear Dr. Rashid:

I'm pleased to inform you that your manuscript has been deemed suitable for publication in PLOS ONE. Congratulations! Your manuscript is now with our production department. 

Kind regards, 

on behalf of

Dr. Charles A. Ameh 

Academic Editor

PLOS ONE